# Disentangling the Janus-faced effects of cations in electrocatalysis

Xinwei Zhu [1,2] ✉, Tobias Binninger [1], Marc T. M. Koper[3] & Michael Eikerling [1,2] ✉

Cation identity and concentration strongly influence electrocatalytic processes, yet their effects remain insufficiently understood. Taking hydrogen evolution reaction in alkaline media as a model system, variations in cation concentration induce complex, sometimes inverted, activity trends. Increasing cation concentration can either promote or inhibit electrocatalytic activity depending on cation identity, electrode material and solution pH. These Janus-faced effects of cations challenge the current understandings of cation effects in electrocatalysis, which typically emphasize either promotional or inhibitory roles. Here, we propose a mechanistic rationale for the promoter-inhibitor transitions of cation effects and identify cation position in the electric double layer as the key factor governing this behavior. The theoretical framework distinguishes two cation states: cations electrostatically attracted in the diffuse layer, or cations specifically adsorbed at the inner Helmholtz plane. Incorporating the electric field effect on water dissociation beyond the Frumkin corrections, we show that the two cation states modulate the local electric field and thus kinetics in opposite ways. The observed inversions result from their competition, governed by cation size and adsorption strength. The framework and insights will be relevant to other electrocatalytic reactions at strongly negatively charged surfaces, such as $CO_2$ reduction.

The critical role of cations in electrocatalytic reactions has been widely recognized since Frumkin's seminal work from 1933[1]. Subsequent studies have demonstrated that both the identity and concentration of cations in the electrolyte influence the kinetics of a broad variety of electrocatalytic processes, including $S_2O_8^{2-}$ reduction[2], hydrogen evolution/oxidation reaction (HER/HOR)[3–7], $CO_2$ reduction reaction ($CO_2$RR)[8–10], and oxygen evolution reaction[11–13]. Among these, HER has long served as a model reaction for probing fundamental aspects of interfacial charge-transfer processes due to its relative simplicity and central role in electrochemical kinetics[3,14,15].

The complexity of cation effects is underscored by multiple inversions in cation-dependent activity trends that are induced by changes in electrode material, applied overpotential, and solution pH,

as illustrated in Fig. 1 for HER under alkaline conditions. On Au electrodes, HER activity follows the order $Li^+ < Na^+ < K^+$ at low overpotentials, whereas at more cathodic potentials, the order inverts to $Li^+ > Na^+ > K^+$[7]. On Pt, the activity follows the trend $Li^+ > Na^+ > K^+$ across the relevant potential range[6,16–18].

In addition to these trends that depend on cation identity, activity inversions have also been reported as a function of cation concentration. Recent studies at pH 11 and 13, on various electrocatalysts, reveal markedly different behaviors. For instance, at pH 11, increasing the concentration of supporting cations uniformly enhances HER activity on Au, independent of cation identity. At pH 13, however, this promotion effect persists only for $Li^+$; increasing the concentration of $Na^+$ or $K^+$ instead leads to a suppression of HER activity[7,19,20]. These

[1]Theory and Computation of Energy Materials (IET-3), Institute of Energy Technologies, Forschungszentrum Jülich GmbH, Jülich, Germany. [2]Chair of Theory and Computation of Energy Materials, Faculty of Georesources and Materials Engineering, RWTH Aachen University, Aachen, Germany. [3]Leiden Institute of Chemistry, Leiden University, Leiden, The Netherlands. ✉e-mail: x.zhu.electrochem@gmail.com; m.eikerling@fz-juelich.de

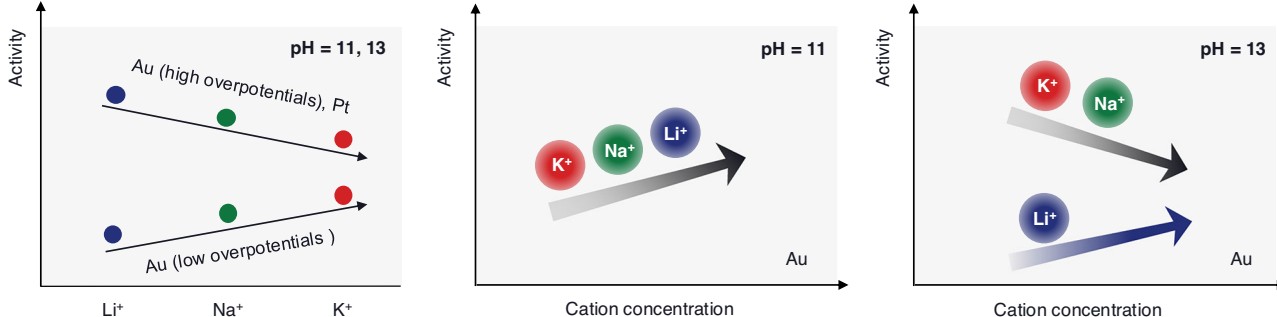

**Fig. 1 | Experimental observations of multiple inversions in cation-dependent HER activity under alkaline conditions.** For cation identity, HER activity on Au follows the trend $Li^+ < Na^+ < K^+$ at low overpotentials, but this trend inverts to $Li^+ > Na^+ > K^+$ at high overpotentials. On Pt, the activity consistently follows the trend $Li^+ > Na^+ > K^+$ across the entire potential range. For cation concentration, increasing the concentration of supporting cations promotes HER activity on Au at pH 11. At pH 13, increasing $Li^+$ concentration enhances HER activity, while increasing $Na^+$ or $K^+$ concentration inhibits it[7,19].

promoter-inhibitor transitions highlight the Janus-faced effects of cations.

Various mechanisms have been proposed to explain these diverse observations, with particular attention focused on the influence of cation identity. One widely discussed explanation is the hydrogen binding energy (HBE) mechanism, which postulates that the strength of hydrogen adsorption varies with cation species, following the trend $Li^+ < Na^+ < K^{+[6]}$. Within this framework, the inverted HER activity trends observed on Au ($Li^+ < Na^+ < K^+$) and Pt ($Li^+ > Na^+ > K^+$) are interpreted via the Sabatier principle, *i.e.*, hydrogen adsorption is too weak on Au but too strong on Pt, with cations modulating the adsorption strength toward or away from the optimal value[6]. However, this explanation is challenged by first-principles calculations, which indicate that the HBE is largely insensitive to the identity of the alkali metal cations[21]. Furthermore, the experimentally observed cation dependence of hydrogen adsorption has been attributed to the co-adsorption of cations and hydroxyl species, rather than to intrinsic changes in the HBE[22].

An alternative line of reasoning links cation identity to the interfacial environment. Two main mechanisms have been proposed under this reasoning. The first invokes differences in solvent reorganization energy to explain the activity trend on Pt[17]. It is hypothesized that solvent reorganization energy increases in the order $Li^+ < Na^+ < K^+$, leading to $Li^+ > Na^+ > K^+$ in HER activity as predicted by Marcus theory[23]. However, this mechanism fails to account for the trend reversal observed on Au, *i.e.*, $Li^+ < Na^+ < K^+$ at low overpotentials.

The second mechanism assumes that $OH^-$ transport from the interface is rate-determining on Pt, and that cations suppress this transport with increasing strength in the order $Li^+ < Na^+ < K^{+[16,24,25]}$. The H-bond network mechanism, which ascribes cation effects to modulation of interfacial H-bond network connectivity and thereby proton transfer, follows a similar rationale and can be regarded as part of the same mechanistic family[21,26,27]. Despite its intuitive appeal, this explanation raises concerns. Firstly, if $OH^-$ transport is the rate-determining step, one would expect stronger inhibition at more negative potentials as the electric double layer (EDL) becomes increasingly rigid[26,28]. Yet, experimentally, the current increases with increasing cathodic potential, contradicting this expectation. Secondly, this mechanism does not account for the observed enhancement in HER activity upon increasing the concentration of cations[7,15,19], which should strengthen, rather than alleviate, the inhibition of $OH^-$ transport[27]. Furthermore, recent studies have demonstrated that the key to enhancing HER activity in alkaline media is the destabilization of the H−OH bond in interfacial water, a view that is inconsistent with the $OH^-$ transport limitation mechanism[29–31].

Importantly, the mechanisms discussed above focus exclusively on cation identity effects, while the origins of cation concentration effects remain poorly understood[20]. Taken together, it is fair to say that

the role of cations in governing alkaline HER kinetics is still only partially resolved, particularly regarding the multiple inversions of both cation identity- and concentration-dependent activity trends.

In this work, we propose a conceptual framework that can disentangle the Janus-faced cation effects. This framework distinguishes two distinct populations of cations present at strongly charged interfaces under alkaline conditions: (i) partially desolvated cations specifically adsorbed at the inner Helmholtz plane (IHP), and (ii) solvated cations concentrated by electrostatic attraction within the diffuse layer. These two populations influence HER kinetics by modulating the local electric field, which plays a key role in activating the bond breaking of reactive water molecules. Importantly, the main premise of our model is that the two populations incur opposing impacts on HER kinetics, *i.e.*, specifically adsorbed cations tend to inhibit activity, while diffuse-layer cations promote it. The multiple inversions are rationalized as the result of competition between these two opposing contributions. Within this framework, the experimentally observed inversions described above can be qualitatively explained.

## Results

Frumkin corrections are the starting point of our analysis, because they capture several experimental trends in cation-dependent HER activity. However, their inability to account for the observed inversions reveals their limitations. To address this, we refine the theoretical framework by incorporating the specific adsorption of partially desolvated cations, and the influence of local electric field on water dissociation. The refined framework is then used to decipher the complex cation effects on HER kinetics.

### Cation effects from the view of Frumkin corrections

We focus on the Volmer step, which is generally regarded as the rate-determining step (RDS) of alkaline HER for the catalysts that bind hydrogen weakly, such as Hg, Ga and Au[3,14,32]. This step corresponds to the discharge of water molecule in alkaline media,

$$H_2O + e^- \rightarrow H_{ads} + OH^-$$

Frumkin corrections underscore the importance of the local potential at the reaction plane, which governs the local concentrations of reactive species and the thermodynamic driving force. According to the Butler-Volmer-Frumkin theory, the corrected Tafel relation of this step is given by [4,33],

$$\log|j| = \frac{\alpha F}{RT\ln 10}\eta + \frac{\alpha F}{RT\ln 10}\psi_x + \text{const} \tag{1}$$

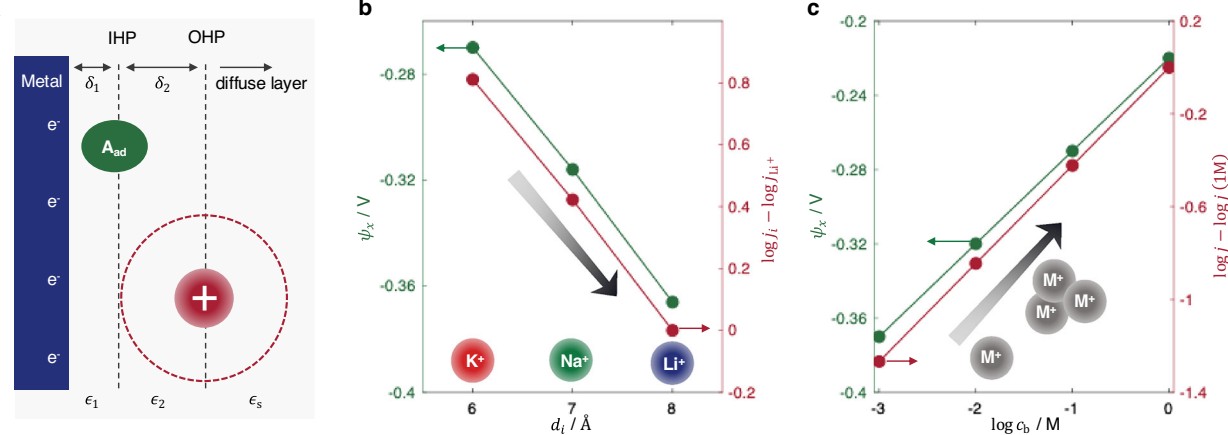

**Fig. 2 | Cation effects from the view of Frumkin corrections. a** Schematic illustration of the conventional EDL structure with nonspecifically adsorbed cations. The continuum EDL model includes the inner Helmholtz plane (IHP), representing the central plane of chemisorbed species, and the outer Helmholtz plane (OHP), representing the central plane of nonspecifically adsorbed cations. The region between the metal surface and the IHP, referred to as the inner Helmholtz layer (IHL), and the region between the IHP and the OHP, referred to as the outer Helmholtz layer (OHL), are described as dielectric continua with respective thicknesses $\delta_i$ and permittivities $\epsilon_i$. **b** Dependence of $\psi_x$ (assumed $\psi_x = \psi_{OHP}$) and HER rate on effective cation size, using $c_b = 0.1$ M. **c** Dependence of $\psi_x$ and HER rate on supporting cation concentration, using $d_i = 6$ Å. The results are calculated using Eqs. 1 and 2. $\alpha$ is set to the typical value of 0.5. $E_M = -0.6$ V versus the reversible hydrogen electrode (RHE) and pH = 13. Other parameters are given in Table 1. The simulated trends do not change with the position of reaction plane, as shown in Supplementary Fig. S1 for $\psi_x = \psi_{IHP}$. Source data are provided as a Source Data file.

with $\eta$ being the overpotential (defined as positive), $j$ the current density, $\alpha$ the transfer coefficient, $\psi_x$ the local potential at the reaction plane, $T$ the temperature (taken as 298 K in this study), $F$ the Faraday constant, and $R$ the ideal gas constant. Here, only the forward reaction is considered, following the RDS assumption. The concentration of water is assumed to remain constant, in line with experimental expectation and with first-principles calculations[26,34,35].

Equation 1 reveals that a more positive $\psi_x$ is expected to enhance HER in alkaline solutions. Within this framework, the role of cations in the supporting electrolyte arises from their influence on the spatial potential distribution across the EDL. Specifically, variations in cation identity and concentration modulate the EDL structure and thereby alter $\psi_x$, ultimately impacting HER kinetics.

$\psi_x$ can be quantitatively evaluated using the modified Gouy–Chapman–Stern (GCS) theory, with finite size effects described by the Bikerman model[36]. The classic EDL structure is illustrated in Fig. 2a, comprising multiple discrete layers characterized by discontinuities in dielectric permittivity. For systems containing monovalent electrolytes, the EDL model is constructed by partitioning the potential drop from the metal surface to the bulk solution[37,38].

$$E_M - E_{pzc} = \sigma_M \left( \frac{\delta_1}{\epsilon_1} + \frac{\delta_2}{\epsilon_2} \right) + \text{sign}(\sigma_M) \frac{2RT}{F} \text{arsinh}\left( \sqrt{\frac{1}{2\gamma} \left( \exp\left( \frac{\gamma}{2} \left( \frac{F\lambda_D \sigma_M}{RT\epsilon_s} \right)^2 \right) - 1 \right)} \right)$$

(2)

where $E_M$ and $E_{pzc}$ are the electrode potential and potential of zero charge (PZC) relative to a reference electrode, respectively; $\sigma_M$ is the electrode surface charge density; $\delta_1$ and $\delta_2$ are the thicknesses, and $\epsilon_1$ and $\epsilon_2$ the dielectric permittivities of the inner and outer Helmholtz layers, respectively. The first term on the right-hand side represents the potential drop across the compact Helmholtz layers, while the second term captures the potential drop across the diffuse layer. Here, $\epsilon_s$ is the dielectric permittivity of electrolyte and $\lambda_D = \sqrt{\frac{RT\epsilon_s}{2F^2 c_b}}$ is the Debye length, with $c_b$ being the bulk cation concentration. $\gamma = 2c_b d_i^3$ accounts for finite ion size effects, where $d_i$ is the effective ion size, defined as a phenomenological lattice cell size that determines the excluded-volume entropy and the maximum local ion concentration.

Cations are distinguished by their effective sizes. Although reported values vary considerably across different sources, the relative trend is generally consistent, following the order $d_{Li^+} > d_{Na^+} > d_{K^+}$, due to the stronger solvation of Li$^+$ compared to Na$^+$ and K$^{+[9,13,39-42]}$. Importantly, this trend is preserved at charged interfaces, as supported by first-principles calculations[10,43], semiclassical modeling[44], and spectroscopic studies[9,45]. Accordingly, we assume representative values of $d_{Li^+} = 8$ Å, $d_{Na^+} = 7$ Å, and $d_{K^+} = 6$ Å to reflect this trend. It should be noted that variations in the exact values do not affect the qualitative conclusions of the analysis below.

In the case of $E_M - E_{pzc} < 0$, $\psi_x$ is negative. As the effective cation size increases, $\psi_x$ becomes more negative, as illustrated in Fig. 2b. According to Eq. 1, a more negative $\psi_x$ slows down HER activity. Therefore, HER activity decreases with effective cation size, following the trend K$^+$ > Na$^+$ > Li$^+$. This prediction aligns with experimental observations on Au at low overpotentials (Fig. 1). However, it fails to capture the inverted trend (Li$^+$ > Na$^+$ > K$^+$) observed on Au at high overpotentials and on Pt across the entire potential range.

Similarly, as the concentration of supporting cations increases, $\psi_x$ becomes less negative, as shown in Fig. 2c, leading to an enhancement in HER activity. This trend is consistent with experimental results for all cations at pH 11 on Au. However, at pH 13, while Li$^+$ still promotes HER at higher concentrations, increasing Na$^+$ or K$^+$ concentrations results in activity suppression, an opposite trend that is illustrated schematically in Fig. 1.

These qualitative discrepancies between theoretical predictions and experimental observations suggest that additional EDL effects, beyond those captured by the GCS model and Butler-Volmer-Frumkin theory, are at play under alkaline conditions.

## Refinement of the EDL model

In alkaline media, the electrode surface is strongly negatively charged due to a positive shift of the PZC on the reversible hydrogen electrode (RHE) scale[28]. For example, in the case of HER on Au in a solution at pH = 13, the potential range of interest is typically more negative than −0.5 V versus RHE. Given that the PZC of Au(111) is -0.5 V versus the standard hydrogen electrode (SHE)[46], and assuming that the PZC is pH-

**a**

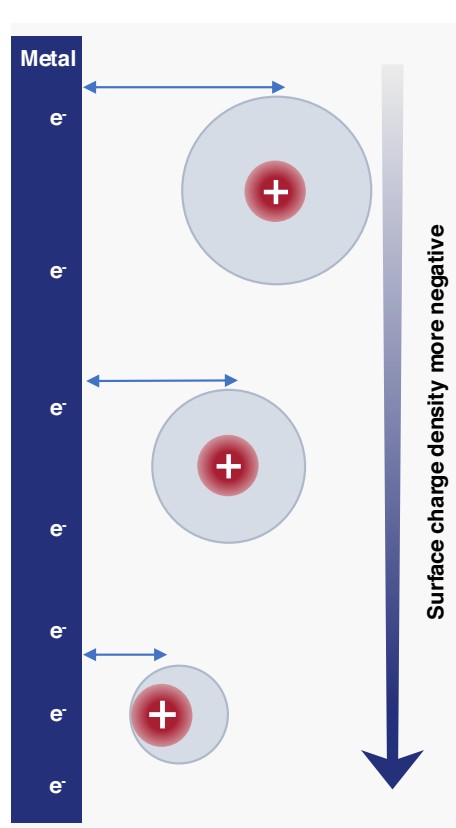

**b**

**c**

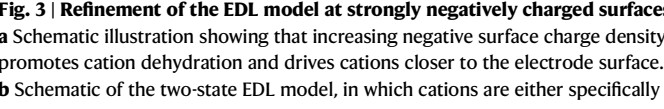

**Fig. 3 | Refinement of the EDL model at strongly negatively charged surfaces.** **a** Schematic illustration showing that increasing negative surface charge density promotes cation dehydration and drives cations closer to the electrode surface. **b** Schematic of the two-state EDL model, in which cations are either specifically adsorbed at the IHP or remain electrostatically attracted in the diffuse layer. **c** Schematic comparison of the potential distribution across the EDL with (II) and without (I) specific cation adsorption.

independent on the SHE scale[47], this corresponds to $E_M - E_{pzc} < -1.8\,V$.

We expect that the more negative surface charge density in alkaline conditions leads to a stronger interfacial electric field, which in turn promotes the partial desolvation of hydrated cations, as illustrated in Fig. 3a [44]. As a result, cations can approach closer to the catalyst surface. In limiting cases, they may penetrate the first water layer such that no water molecules reside between the cations and the electrode surface. These cations are classified as specifically adsorbed cations[48-50].

Since these cation states reshape the potential distribution across the EDL[51], a refined EDL model accounting for these effects is required. To this end, we propose a simplified two-state EDL model, in which cations exist in one of two states: (i) specifically adsorbed at the IHP, or (ii) electrostatically attracted and located outside the OHP, as schematically illustrated in Fig. 3b. Intermediate configurations are not considered here, as they correspond to a shift in the position of OHP (Fig. 2) and do not qualitatively change the structure of the EDL[45].

The specifically adsorbed cations form a positive charge layer at the IHP, causing an additional potential rise across the EDL, Fig. 3c. The modified EDL model that incorporates this influence is given by[37,52,53],

$$E_M - E_{pzc} = (\sigma_{free} - \sigma_{ad})\frac{\delta_1}{\epsilon_1} + \sigma_{free}\frac{\delta_2}{\epsilon_2}$$
$$+ \text{sign}(\sigma_{free})\frac{2RT}{F}\text{arsinh}\left(\sqrt{\frac{1}{2\gamma}\left(\exp\left(\frac{\gamma}{2}\left(\frac{F\lambda_D\sigma_{free}}{RT\epsilon_S}\right)^2\right) - 1\right)}\right)$$

(3)

where $\sigma_{free} = \sigma_M + \sigma_{ad}$ is the free surface charge density, balanced by the charge in the diffuse layer (i.e., the electrostatically attracted cations), and $\sigma_{ad} = e_0\rho\xi_i\theta_i$ represents the contribution from specifically adsorbed cations[54]. Here, $\xi_i$ is the charge number per adsorbed cation and can be estimated from, e.g., Bader charge analysis, $\theta_i$ is the surface coverage, $e_0$ is the elementary charge, and $\rho$ is the number density of adsorption sites on the catalyst surface.

The cation adsorption, $M^+ + (1 - \xi_i)e^- \rightleftharpoons M_{ad}^{+\xi_i}$, can be described using the Frumkin isotherm[55],

$$\ln\left(\frac{\theta_i}{\theta_{max} - \theta_i}\right) + \gamma_i\theta_i = -\frac{(1 - \xi_i)e_0\left(E_M - E_i^0\right)}{k_B T} + \ln c_i \quad (4)$$

where $\theta_{max}$ is the maximum coverage, $\gamma_i$ is the lateral interaction coefficient, $c_i$ is the cation concentration, $E_i^0$ is the standard equilibrium potential of cation adsorption, and $k_B$ the Boltzmann constant. We note that this is a simplified treatment. A more comprehensive description would require accounting for the complex interactions between cations, water molecules, and the catalyst surface, and would relax the mean-field description[56]. Nevertheless, it qualitatively captures two key characteristics, i.e., cation coverage increases with more negative applied potential and with higher cation concentration[50].

Like cations at the OHP, specifically adsorbed cations in the IHP shift $\psi_x$ to more positive values, which promotes HER kinetics according to the Frumkin corrections (Eq. 1), as shown in Fig. 4a. It is reasonable to assume that the coverage of adsorbed cations increases with bulk cation concentration. This implies that the promotional

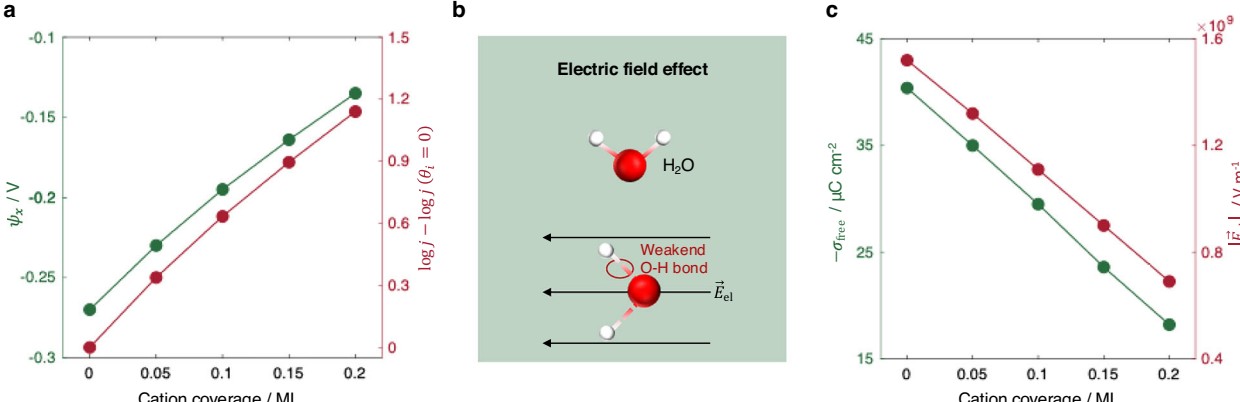

**Fig. 4 | Cation adsorption effects on local potential and electric field. a** Influence of the coverage of specifically adsorbed cations on $\psi_x$ (assumed $\psi_x = \psi_{OHP}$) and HER rate. **b** Schematic of the electric field effect on water dissociation, wherein the electric field weakens the H-OH bond strength. **c** Influence of the coverage of specifically adsorbed cations on $\sigma_{free}$ and the resulting electric field strength within the OHL. The results are calculated using Eqs. 1, 3 and 5. The conditions used in the simulations are: $d = 6\,Å$, $c_b = 0.1\,M$, $E_M = -0.6\,V$ versus RHE, and pH = 13. Other parameters are given in Table 1. Source data are provided as a Source Data file.

effect of increasing cation concentration persists qualitatively in the presence of specific adsorption. In other words, the trend obtained in Fig. 2c remains qualitatively valid. Specific adsorption introduces only a quantitative correction rather than a qualitative inversion. Consequently, the experimentally observed suppression of HER activity upon increasing Na$^+$ or K$^+$ concentration remains unexplained. This discrepancy motivates us to introduce additional EDL effects beyond the Butler-Volmer-Frumkin theory.

### Refinement of the model for electron transfer kinetics

Compared to considering the HER under acidic conditions, reactant species under alkaline conditions change from protons to water molecules, implying that an additional energy penalty occurs due to the need to break the H-OH bond and initiate water dissociation[29]. It has long been recognized that an external electric field can facilitate electrolytic dissociation, a phenomenon known as the second Wien effect[57–59]. This field-assisted water dissociation, illustrated schematically in Fig. 4b, has been supported by experimental observations[60–63] as well as theoretical studies[64,65]. Factors that suppress the interfacial electric field strength will thus impede water dissociation and thereby inhibit HER kinetics.

We, therefore, proceed to examine the influence of cation adsorption on the electric field strength within the outer Helmholtz layer (OHL), the region between IHP and OHP, where the reactive water molecules are assumed to reside, as shown in Fig. 3b. This assumption is supported by ab initio molecular dynamics (AIMD) simulations[26], and surface X-ray diffraction[66], which show that the innermost cation layer is integrated with the first interfacial water layer. The electric field strength across this layer is proportional to the free surface charge density and is given by,

$$\left|\vec{E}_{el}\right| = \left|\sigma_{free}\right| / \epsilon_2 \qquad (5)$$

Figure 4c shows that $\left|\sigma_{free}\right|$ decreases with cation coverage, leading to a reduction in the electric field strength within the OHL. This suggests that specific cation adsorption weakens $\left|\vec{E}_{el}\right|$, thereby slowing down water dissociation and, in turn, HER activity.

To quantify this effect, we assume that the H−OH bond strength $D$ decreases linearly with the electric field strength,

$$D = D_0 - B\left|\vec{E}_{el}\right| \qquad (6)$$

with $D_0$ being the H·OH bond strength in the absence of an electric field, and $B$ being a coefficient characterizing the field sensitivity of the bond.

The electron transfer with water bond-breaking is described using a model Hamiltonian-based approach, which yields the activation energy, following refs. 67–70.,

$$\triangle G_a = \frac{(\lambda + D + e_0\eta - e_0\psi_x)^2}{4(\lambda + D)} + \frac{\Delta}{2\pi}\ln\frac{\Delta^2}{(\lambda + D + e_0\eta - e_0\psi_x)^2 + \Delta^2} \qquad (7)$$

where $\lambda$ is the solvent reorganization energy, $\psi_x$ accounts for the Frumkin correction, and $\Delta$ represents the electronic interaction strength between the adsorbed hydrogen and the metal electrode. In the limit $\Delta = 0$ and under equilibrium, $\eta = 0$, Eq. 7 reduces to $\triangle G_a = \frac{\lambda + D_0}{4} - \frac{B}{4}\left|\vec{E}_{el}\right|$, when Frumkin corrections are neglected. This expression closely follows earlier theoretical treatments of the electrochemical Ce(III)/Ce(IV) redox couple[58], and ionization of water−ice adsorbed onto a platinum surface[71].

The current density is then given by[40,55,72],

$$j = -2\kappa_{el}\upsilon_n\exp\left(-\frac{\triangle G_a}{k_B T}\right)e_0\rho(1 - \theta_i) \qquad (8)$$

with $\kappa_{el}$ being the transmission coefficient, and $\upsilon_n$ the nuclear barrier-crossing frequency.

Equations 3−8 constitute a refined theoretical framework for investigating cation effects on the HER under alkaline conditions. In the following sections, we demonstrate that this framework qualitatively captures all the inversion trends presented in Fig. 1.

### Understanding cation identity effects

Different cations exhibit varying tendencies for specific adsorption, following the trend K$^+$ > Na$^+$ > Li$^+$[50,73]. This is attributed to differences in the hydration strength, i.e., K$^+$ is more readily desolvated due to a weaker hydration shell and thus more likely to specifically adsorb onto the electrode surface compared to Na$^+$ and Li$^+$. In the context of the adsorption isotherm, this trend is reflected in the relative values of standard adsorption potentials, viz. $E_{Li^+}^0 < E_{Na^+}^0 < E_{K^+}^0$.

With this consideration, the model captures the inversion of the dependence of the HER activity on Au electrodes on the cation identity. Specifically, the activity trend follows K$^+$ > Na$^+$ > Li$^+$ at $E_M > -0.76\,V$ and is inverted to K$^+$ < Na$^+$ < Li$^+$ at $E_M < -0.76\,V$ versus RHE, as shown in Fig. 5a and b. The calculated cation coverages

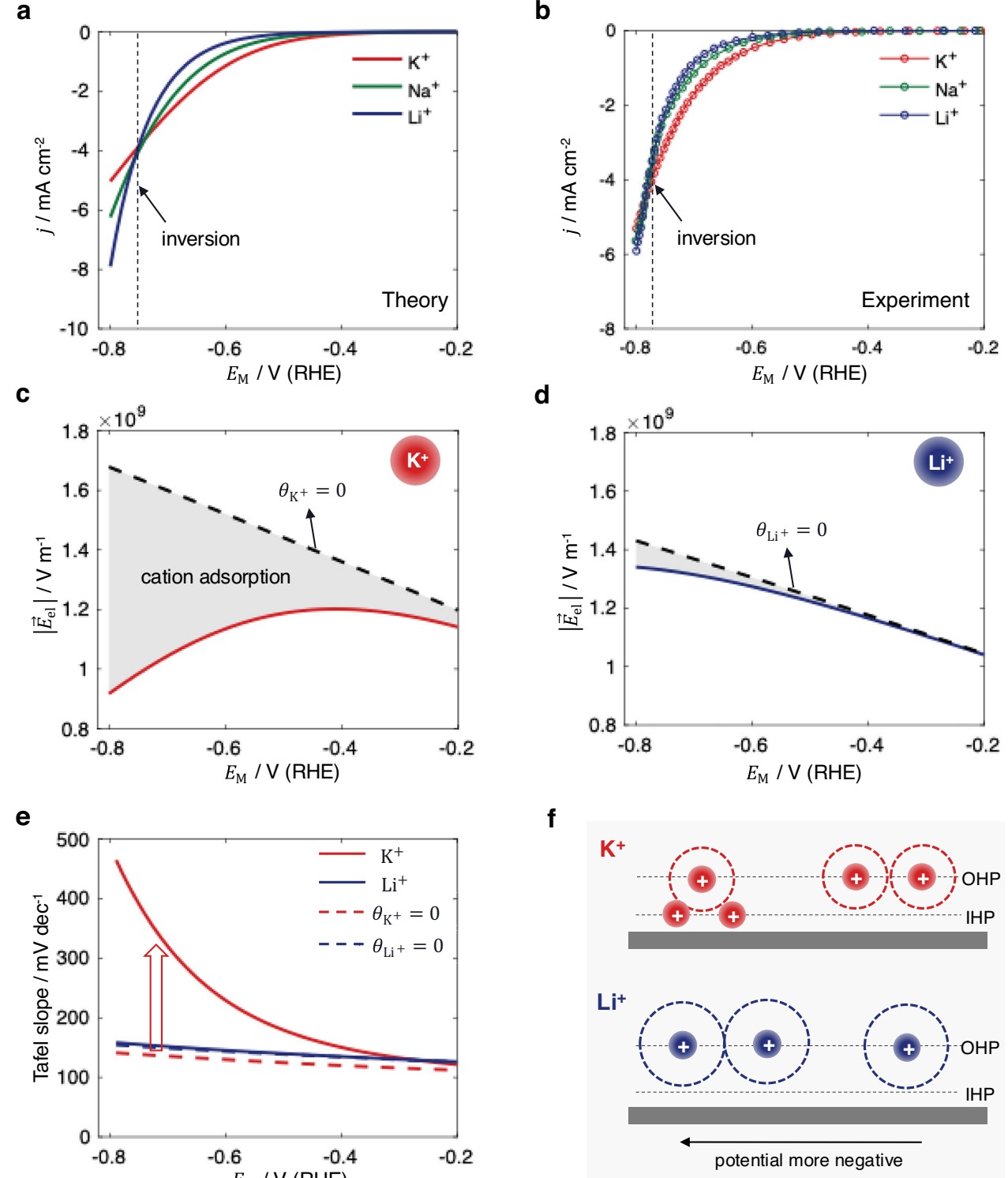

**Fig. 5 | Understanding cation identity effects. a** Simulated and **b** experimental polarizations curves for HER on Au in 0.1 M KOH, NaOH, and LiOH, with experimental data taken from ref.[7]. Calculated electric field strength within the OHL for **c** K$^+$ and **d** Li$^+$. **e** Influence of cation adsorption on the Tafel slopes. The results

without considering cation adsorption are shown for comparison. **f** Schematic illustration of the underlying mechanisms for the inversion of cation identity-dependent activity. The results are simulated using Eqs. 3–8 and parameters in Table 1. Source data are provided as a Source Data file.

follow $\theta_{K^+} > \theta_{Na^+} > \theta_{Li^+}$ and increase with overpotential, Supplementary Fig. S2 in the Supplementary Information (SI), in agreement with the trends observed in spectroscopy studies[50]. The adsorbed cations reduce the electric field strength $|\vec{E}_{el}|$, as illustrated in Fig. 5c and d for $K^+$ and $Li^+$, respectively. This effect is more pronounced for $K^+$ due to its stronger adsorption ability. The results for $Na^+$, representing an intermediate case, are provided in Supplementary Fig. S3.

Consequently, the reduction in H-OH bond strength $D$ with increasing overpotential is attenuated by cation adsorption, as described by Eq. 6 and shown in Supplementary Fig. S2, slowing down HER activity. This effect induces an increase in Tafel slope with overpotential, particularly for $K^+$, as shown in Fig. 5e. Moreover, the model reproduces anomalously high Tafel slopes, in agreement with experimental observations[19]. These high Tafel slopes, which do not appear in simulations that neglect cation adsorption, are attributed to the influence of specifically adsorbed cations, rather than intrinsic HER kinetics. The local potential $\psi_x$, which characterizes the Frumkin corrections, follows the order $K^+ > Na^+ > Li^+$ (Supplementary Fig. S2). This trend is governed primarily by the effective cation size, as analyzed earlier in Fig. 2b.

The underlying mechanism for this inversion is schematically illustrated in Fig. 5f. The cations differ in effective size and adsorption ability in the proposed framework. The effective size trend, $d_{K^+} < d_{Na^+} < d_{Li^+}$, results in the order $K^+ > Na^+ > Li^+$ for $\psi_x$, and HER activity thus follows the same order according to the Frumkin corrections. In contrast, the adsorption ability follows $K^+ > Na^+ > Li^+$, resulting in the trend $\theta_{K^+} > \theta_{Na^+} > \theta_{Li^+}$, which weakens the electric field strength and thus suppresses the HER. Accordingly, the HER activity follows a reversed trend $K^+ < Na^+ < Li^+$ in regimes where adsorption dominates. At moderate overpotentials ($E_M > -0.76$ V), the size effect dominates, while at higher overpotentials ($E_M < -0.76$ V), the adsorption effect prevails. The observed inversion, therefore, originates from the competition between these two opposing factors.

We notice that this inversion on Au at pH 13 was observed in ref. 7., but not in another study[24]. This experimental discrepancy may arise from differences in experimental conditions, such as catalyst preparation methods and electrolyte purification procedures. Within the framework of the proposed mechanism, the occurrence of the inversion depends sensitively on the relative differences in the specific adsorption coverages of the cations. These, in turn, may be influenced by various factors, including metal facets, surface defects, and impurities in the electrolyte solution[15].

For Pt electrodes, the activity consistently follows the trend $K^+ < Na^+ < Li^+$[6,7,24]. A possible explanation is that the cation adsorption ability is the dominant factor across the entire potential range on Pt. This argument is supported by density functional theory (DFT) calculations, which have revealed a more favorable energetics of cation adsorption on Pt surfaces[22,74,75].

## Understanding cation concentration effects

The cation concentration effects on Au are explored at constant overpotential of $E_M = -0.6$ V versus RHE, consistent with experimental conditions[7]. The comparison between theoretical predictions and experimental data at pH 13 is shown in Fig. 6a and b. The model captures two key experimental trends. Firstly, increasing the $Li^+$ concentration promotes HER activity, whereas increasing the $K^+$ or $Na^+$ concentrations suppresses it. Secondly, the reaction order with respect to cation concentration follows the trend $K^+ < Na^+ < Li^+$.

Variations in cation concentration affect both the density of specifically adsorbed cations and electrostatically attracted cations. These two populations correspond to the adsorbate charge density $\sigma_{ad}$ and free surface charge density $\sigma_{free}$, respectively. The metal surface charge density satisfies the relation $|\sigma_M| = |\sigma_{free}| + |\sigma_{ad}|$, considering the overall electroneutrality of the EDL[54]. The surface coverage of specifically adsorbed cations increases with cation concentration and

follows the trend $\theta_{K^+} > \theta_{Na^+} > \theta_{Li^+}$ (Supplementary Fig. S4a). Consequently, $|\sigma_{ad}|$ increases with cation concentration, as shown for $K^+$ and $Li^+$ in Fig. 6c and d, respectively. A similar trend for $Na^+$ is shown in Supplementary Fig. S4b. In contrast, $|\sigma_{free}|$ decreases with increasing cation concentration, with this trend being more pronounced for $K^+$ due to its stronger adsorption tendency.

The metal surface charge density $|\sigma_M|$, representing the combined contribution from both types of cations, increases with cation concentration. This results in a positive shift in the local potential $\psi_x$, as shown in Fig. 6e, promoting the HER activity according to the Frumkin corrections. However, the reduction in $|\sigma_{free}|$ weakens the electric field strength $|\vec{E}_{el}|$, Fig. 6f, thereby slowing down the HER activity.

The underlying mechanism for the promotion or inhibition effects is schematically illustrated in Fig. 6g. Increasing the cation concentration has two competing effects on HER activity: (i) a promotional effect via a positive shift in $\psi_x$, and (ii) an inhibitory effect via a reduction in $|\vec{E}_{el}|$. The net impact depends on which effect dominates. For $Li^+$, the first effect is dominant, leading to enhanced HER activity with increasing $Li^+$ concentration. For $K^+$ and $Na^+$, the second effect dominates, resulting in decreased HER activity upon increasing concentration. This suppression is more pronounced for $K^+$ due to its stronger adsorption.

The situation changes at pH 11, where increasing the concentration of $K^+$ or $Na^+$ also promotes HER activity on Au. The pH-dependent trends are also observed on Pt, i.e., increasing the $K^+$ concentration at pH 9 initially enhances, but subsequently inhibits, HER activity, whereas at pH 11, it consistently suppresses HER activity. These complex behaviors are summarized in Fig. 7a.

We expect that these effects originate from the variations in the specific adsorption of cations, which are influenced by the catalyst material and the solution pH[7,15,22,75]. Therefore, we examine the $K^+$ concentration-dependent activity as a function of $E^0_{K^+}$, which characterizes the adsorption strength of $K^+$. The results, shown in Fig. 7b, display distinct trends. When $E^0_{K^+} = -1.5$ V versus SHE, corresponding to weak adsorption, increasing the $K^+$ concentration promotes the HER activity. When $E^0_{K^+} = -1.2$ V versus SHE, increasing the $K^+$ concentration first promotes, then inhibits the HER activity. When $E^0_{K^+} = -0.9$ V versus SHE, representing strong adsorption, the HER activity is consistently suppressed with increasing $K^+$ concentration.

These varying trends reproduce the complex experimental behaviors in Fig. 7a, where cation adsorption is stronger on Pt and at higher pH[22,74]. We note, however, that HER on Pt involves additional complexities, including strong hydrogen adsorption and co-adsorption of hydroxide species[22,24,76]. Thus, the qualitative comparison in Fig. 7 indicates that the proposed mechanism remains relevant on Pt, but it does not exclude other effects reported in previous studies[15].

## Discussion

We have presented a theoretical framework that explains the complex cation effects in alkaline HER, exhibiting hitherto puzzling trends, by distinguishing two interfacial cation populations, i.e., cations that are electrostatically attracted in the diffuse layer and cations specifically adsorbed at the inner Helmholtz plane. These distributions and their relative proportions modulate the local electric field in opposite ways, giving rise to the promoter−inhibitor transitions of cation effects. The reliability of the proposed model is supported by its ability to rationalize diverse trends in cation identity- and concentration-dependent HER activity across different pH conditions and electrode materials with two key cation properties, i.e., the effective size and the equilibrium potential of specific adsorption.

Although this study focuses on cation effects, the same logic should be applicable to anion effects as well. For instance, a volcano-

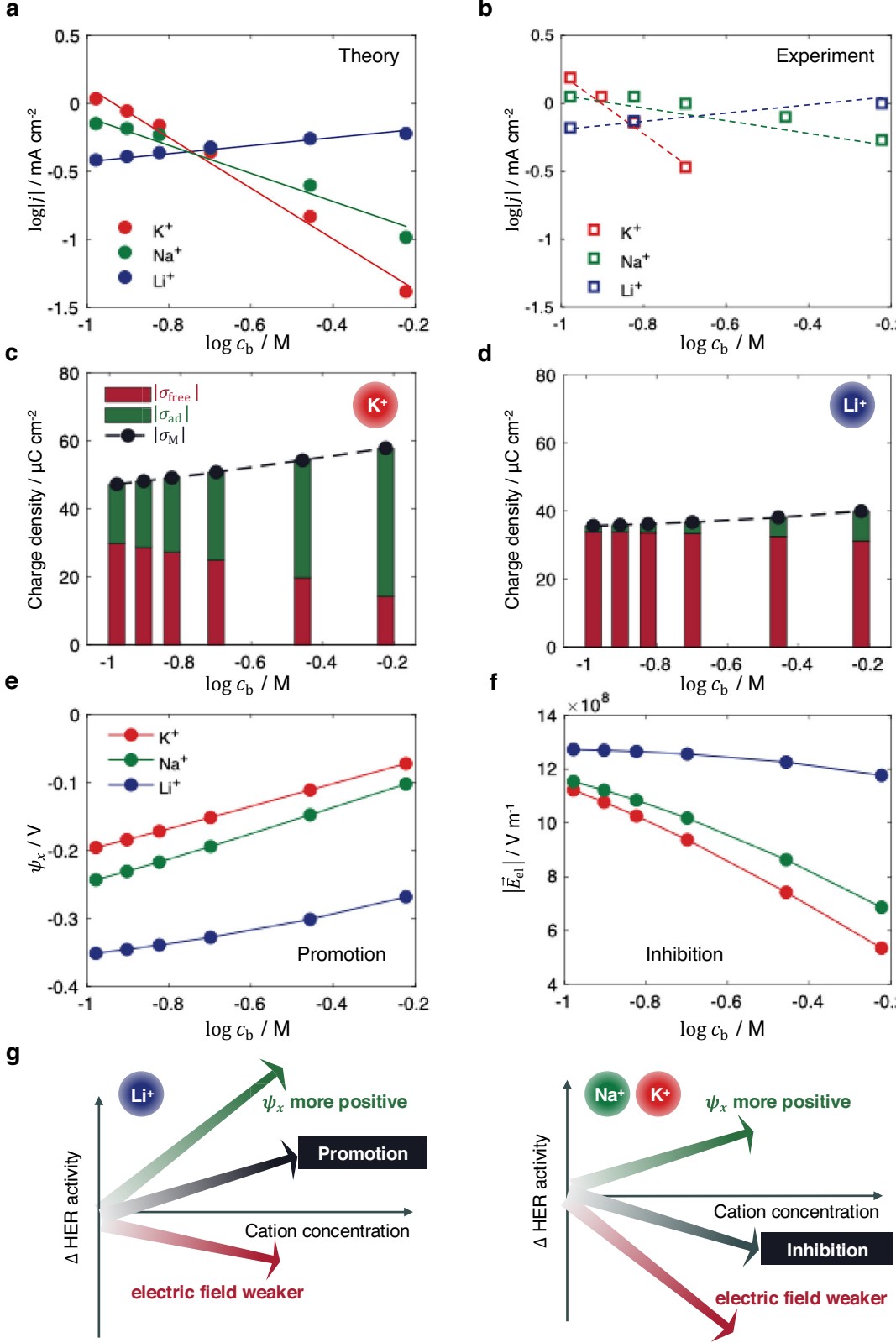

**Fig. 6 | Understanding cation concentration effects. a** Simulated and **b** experimental HER activity on Au with respect to cation concentration, with experimental data taken from refs. 7,19. Influence of cation concentration on surface charge densities for **c** K⁺, and **d** Li⁺. Influence of cation concentration on **e** local potential $\psi_x$ (assumed $\psi_x = \psi_{OHP}$), and **f** electric field strength within the OHL.

**g** Schematic illustration of the underlying mechanisms for the inversion of the cation concentration-dependent activity. Simulation and experimental conditions: $E_M = -0.6$ V versus RHE and pH = 13. The results are simulated using Eqs. 3–8 and parameters in Table 1. Source data are provided as a Source Data file.

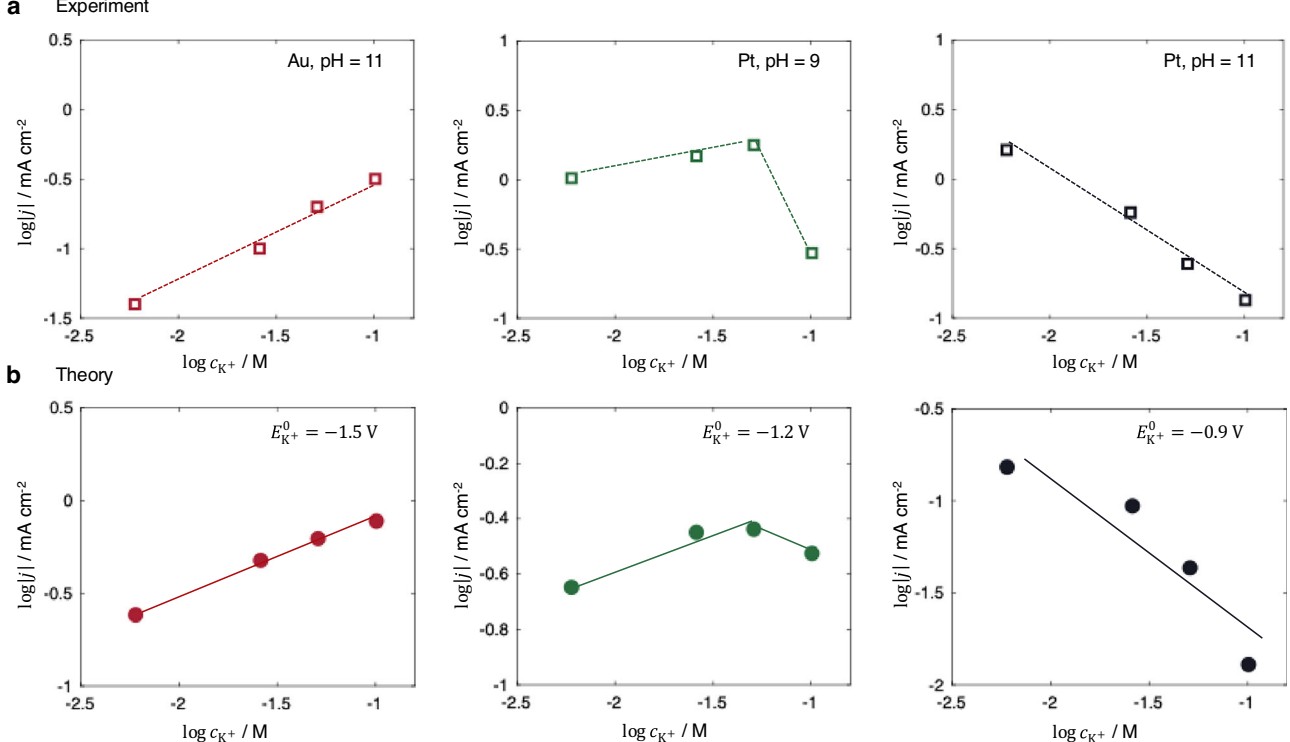

**Fig. 7 | Coupled cation effects with catalyst materials and solution pH.**
**a** Experimental observations showing the influence of $K^+$ concentration on HER activity on Au and Pt electrodes at different pH values. Experimental data are taken from ref. 7. **b** Simulated HER activity as a function of $K^+$ concentration under varying adsorption strengths of $K^+$, while all other parameters remain unchanged. Experimental conditions: $E_M = -0.6$ V versus RHE for Au and $E_M = -0.3$ V versus RHE for Pt. Simulation conditions: $E_M = -0.6$ V versus RHE and pH = 11. Source data are provided as a Source Data file.

type relationship between HER activity and hydroxide binding strength has been reported for modified Pt catalysts[31]. Within our framework, such adsorbed hydroxide species have two opposing effects: (i) enhancing the interfacial electric field due to their residual negative charge[37,52], thereby promoting HER, an effect opposite to that of specifically adsorbed cations; (ii) blocking active sites at high coverage, which inhibits HER. The competition between these two effects provides a qualitative explanation for the observed volcano-type trend.

Beyond HER, this conceptual approach may be relevant for other electrochemical reactions that occur at strongly negatively charged interfaces, such as $CO_2RR$. For cation effects in $CO_2RR$, several mechanisms have been proposed, including effects of cation size[9,77], electric field[78], and the specific adsorption[50]. In experiments, the activity trends of different products with respect to cation identity often appear convoluted[10]. From the perspective developed in this study, such complexity may stem from the simultaneous and competing influences of cations in different interfacial states, paralleling the rationale proposed here for HER.

Insights gained from this work will have implications for optimizing electrolytes for water electrolysis. In practice, highly concentrated KOH is commonly used in alkaline water electrolysis due to its favorable ionic conductivity[79]. However, the presented analysis suggests that specific adsorption of $K^+$ can inhibit HER kinetics, especially at high concentrations. To address this trade-off, mixed-cation electrolytes, such as combinations of CsOH and LiOH[80], may offer a promising solution by retaining high conductivity while mitigating inhibitory adsorption effects. Nonetheless, a deeper theoretical understanding and clear design guidelines for such mixed-cation systems remain to be pursued in future studies.

## Methods
### Model parameterization
The model contains three groups of parameters as summarized in Table 1. The first group relates to the multilayer structure and dielectric properties of the EDL. The second group describes the properties of cations and their specific adsorption. The third group characterizes the kinetics of HER.

The first group of parameters includes $\epsilon_i$, $\delta_i$, and $E_{pzc}$. For the inner Helmholtz layer, we set $\delta_1 = 2$ Å, which places it slightly closer to the metal surface than the first-layer water molecules[35]. $\epsilon_1 = 8\epsilon_0$ is higher than the vacuum permittivity to account for the electron spill-over of the metal[37]. For the outer Helmholtz layer, we set $\delta_2$ being the radius of hydrated cations, i.e., $\delta_2 = d_i/2$; $\epsilon_2 = 30\epsilon_0$, as estimated by Bockris and coworkers[81]. $\epsilon_s = 78\epsilon_0$ is the permittivity of bulk solution. $E_{pzc} = 0.5$ V versus the SHE for Au(111)[46].

The second group of parameters includes $d_i$, $\xi_i$, $\gamma_i$, and $E_i^0$. While the effective size $d_i$ varies significantly across literature sources, the trend is generally consistent, i.e., $d_{Li^+} > d_{Na^+} > d_{K^+}$ [9,13,39,40]. We assume $d_{Li^+} = 8$ Å, $d_{Na^+} = 7$ Å, and $d_{K^+} = 6$ Å to reflect this trend. The resulting distances between the cations and the electrode surface, i.e., $\delta_1 + \delta_2$, lie between 5 Å and 6 Å, consistent with molecular dynamic simulations[82,83]. Recent AIMD studies combined with Bader charge analysis indicate that $\xi_i$ is only weakly dependent on cation identity, with typical values obtained in the range 0.8-0.9 for alkali metal cations[43,50,83]. We thus adopt $\xi_i = 0.85$ for all cations. The sensitivity of model results to $\xi_i$ has been assessed by performing a sensitivity analysis as reported in Supplementary Fig. S5. $\gamma_i$, $\theta_{max}$, and $E_i^0$ are fitted to the experimental polarization curves in Fig. 5b. $\gamma_i = 3$ accounts for the repulsion between adsorbed cations. The maximum surface coverage of specifically adsorbed cations is constrained $\theta_{max} = 0.5$. The

## Table 1 | Model parameters

| Category | Item | Value |
|---|---|---|
| EDL parameters | Thickness of IHL, $\delta_1$ | 2 A |
| | Thickness of OHL, $\delta_2$ | $d_i/2$ |
| | Permittivity of IHL, $\epsilon_1$ | $8\epsilon_0$ |
| | Permittivity of OHL, $\epsilon_2$ | $30\epsilon_0$ |
| | Permittivity of bulk solution, $\epsilon_s$ | $78\epsilon_0$ |
| | Potential of zero charge, $E_{pzc}$ | 0.5 V vs. SHE for Au(111) |
| Cation properties | Effective diameter of Li⁺, $d_{Li^+}$ | 8 A |
| | Effective diameter of Na⁺, $d_{Na^+}$ | 7 A |
| | Effective diameter of K⁺, $d_{K^+}$ | 6 A |
| | Charge number of per adsorbed cation, $\xi_i$ | 0.85 for Li⁺, Na⁺, K⁺ |
| | Lateral interaction coefficient, $\gamma_i$ | 3 for Li⁺, Na⁺, K⁺ |
| | Adsorption equilibrium potential, $E^0_{Li^+}$ | −1.65 V versus SHE |
| | Adsorption equilibrium potential, $E^0_{Na^+}$ | −1.27 V versus SHE |
| | Adsorption equilibrium potential, $E^0_{K^+}$ | −1.19 V versus SHE |
| | Maximum coverage, $\theta_{max}$ | 0.5 |
| Kinetic parameters | Solvent reorganization energy, $\lambda$ | 3 eV |
| | Field-free H-OH bond strength, $D_0$ | 4.7 eV |
| | Coefficient for electric field effects, $B$ | $1.7 \times 10^{-9} e_0$ m |
| | Equilibrium potential of Volmer step, $E_{eq}$ | −1.24 V versus SHE |
| | Electronic interaction strength, $\Delta$ | 3 eV |
| | Active site density, $\rho$ | 0.14 A$^{-2}$ |
| | Transmission coefficient, $\kappa_{el}$ | 1 |
| | Nuclear barrier-crossing frequency, $v_n$ | $\frac{k_B T}{h}$ |

cation adsorption ability follows the trend K⁺ > Na⁺ > Li⁺[50,51], implying $E^0_{Li^+} < E^0_{Na^+} < E^0_{K^+}$. We obtained $E^0_{Li^+} = -1.65$ V, $E^0_{Na^+} = -1.27$ V, and $E^0_{K^+} = -1.19$ V versus SHE by fitting to experimental polarization curves. The validity of these values is supported by the results in Fig. 6, where the same set of parameters reproduces also the experimental trends for cation concentration effects.

The third group of parameters includes $\lambda$, $D$, $\eta$, $\Delta$, and $\rho$. $\lambda = 3$ eV is taken from molecular dynamics simulations[84], and is consistent with earlier estimates of Schmickler and coworkers[70]. $D$ is calculated using Eq. 6, with $D_0 = 4.7$ eV being the H-OH bond strength without electric field influence[85], and $B = 1.7 \times 10^{-9} e_0$ m being a fitted parameter. The calculated $D$ is significantly lower than $D_0$ and close to the estimated value in a previous study (about 2 eV), as shown in Supplementary Fig. S2[69]. $\eta = E_M - E_{eq}$, where $E_{eq} = -1.24$ V versus SHE is the standard equilibrium potential of the alkaline Volmer step on Au. This value is derived from $E_{eq} = -\triangle G^0/e_0$, with $\triangle G^0 = 1.24$ eV being the reaction free energy based on the standard hydrogen adsorption free energy of $\triangle G^0_H = 0.41$ eV[70,86,87]. $\Delta$ is assumed constant under wide-band approximation, and $\Delta = 3$ eV is used according to DFT calculations[70]. $\rho = 0.14$ A$^{-2}$ for Au(111) based on a lattice constant of 4.08 A[75]. The transmission coefficient $\kappa_{el}$ is set to 1, as expected for adiabatic electron transfer step[55]. The nuclear barrier-crossing frequency is assumed to be $v_n = \frac{k_B T}{h}$, with $h$ being Planck constant[55].

## Numerical solution

The models are solved using MATLAB, and all the codes are provided in the SI. Using these codes, the main figures can be reproduced.

## Parameter sensitivity analysis

Cation specificity in our framework is captured through two physically motivated parameters with well-established trends, i.e., the effective size, $d_i$, and the equilibrium cation adsorption potential, $E^0_i$, while all other parameters are kept identical for different cations. This strategy maintains a minimal and physically interpretable parameter set and avoids overfitting.

To disentangle the influence of other parameters, including $\xi_i$, $\gamma_i$, $\epsilon_1$ and $\epsilon_2$, we performed sensitivity analyses, in which these parameters were independently varied over reasonable ranges. The impact of these variations on key observables, including cation adsorption coverage, EDL characteristics ($\psi_x$ and $|\vec{E}_{el}|$), and HER current density, has been systematically analyzed.

The results, summarized and discussed in Supplementary Note 1 in the SI, demonstrate that variations in these parameters affect the magnitude of interfacial potential, electric field strength, and HER current density, but do not reproduce the experimentally observed inversion of cation trends without accounting for the systematic trends in effective cation size and adsorption strength. This supports the proposed mechanism that attributes the inversions of cation effects to the competition between Frumkin effect and electric field effect, both of which are ultimately governed by cation size and adsorption strength.

While we believe that the conclusions are well justified and robust, we expect our work to stimulate more detailed simulations at the level of DFT and AIMD to systematically scrutinize the impact of different cation states on the HER kinetics.

## Data availability

Source data underlying the figures are provided with this paper. All data supporting the findings of this study are generated from the theoretical models described in the manuscript. Source data are provided with this paper.

## Code availability

The MATLAB codes used to perform the calculations and generate the figures are provided in the Supplementary Information.

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

## Acknowledgements

X.Z., T.B., and M.E. acknowledge funding from the Helmholtz-Gemeinschaft Deutscher Forschungszentren e.V. (HGF), Program-oriented Funding (PoF IV), under the Research Program: Materials and Technologies for the Energy Transition (MTET). This publication is part of the DECODE project that has received funding from the European Union's Horizon Europe research and innovation programme under grant agreement No. 101135537. M.T.M.K. acknowledges funding by the European Research Council (ERC), Advanced Grant No. 101019998 "FRUMKIN". Views and opinions expressed are however those of the author(s) only and do not necessarily reflect those of the European Union or HADEA. Neither the European Union nor the granting authority can be held responsible for them. We thank Prof. Olaf Magnussen and Dr. Arthur Hagopian for insightful discussions. X.Z. gratefully acknowledges Prof. Jun Huang for valuable suggestions on the writing and conceptual development of this work.

## Author contributions

X.Z., M.T.M.K., and M.E. conceived the project. X.Z. performed the modeling and wrote the manuscript. T.B. assisted in the theoretical analysis. All authors discussed the results, commented on the manuscript, and approved the final version of the manuscript.

## Funding

## Competing interests

The authors declare no competing interests.
