## [Transparent Peer Review file · Nature Communications]

Disentangling the Janus-faced effects of cations in electrocatalysis

Corresponding Author: Mr Xinwei Zhu

Version 0:

Reviewer comments:

Reviewer #1

(Remarks to the Author)

This work took hydrogen evolution reaction (HER) in alkaline media as a representative model system and revisited the electric double layer (EDL) model and electron-transfer kinetics by considering impact of cation position on local electric field. They revealed that cations electrostatically adsorbed in the diffuse layer and those specifically adsorbed at the inner Helmholtz plane would have modulate the HER kinetics in opposite way: the former enhances HER while the latter suppresses it. The Janus-faced cation effects disclosed by their theoretical model successfully explained the observed manifold of cation effects in HER on Pt and Au. Overall, this is a timely and well-crafted paper that provides new physical insight into cations' influences in EDL formed on negatively charged catalyst surface. Specific comments to be addressed before further consideration are as follows:

1. Previous studies (e.g., Phys. Chem. Chem. Phys., 2024, 26, 18233–18243; Nature Energy 5, 891–899 (2020)) have reported that under alkaline conditions there can be a relatively high OH/O coverage on metal surfaces. If OH/O adsorption is considered, qualitatively, how would it affect the conclusions of the present work?
2. At the beginning of the manuscript, the authors state: "We focus on the Volmer step, which is generally regarded as the rate-determining step (RDS) of alkaline HER for the catalysts that bind hydrogen weakly, such as Hg, Ga and Au." However, Pt is known to bind hydrogen relatively strongly (ACS Energy Lett. 2023, 8, 657–665). Why does the improved model proposed in this work also successfully fit and explain the experimental results for Pt, a metal with strong hydrogen adsorption?
3. In the approximation treatment, two parameters— ψ_x (the local potential at the reaction plane, where a more positive ψ is expected to enhance HER in alkaline solutions) and the electric field—are considered separately. If the author can prove that the assumption holds in realistic conditions? If these two variables are associated, it would be necessary to discuss to what extent this approximation of treating them independently may affect the results.
4. Can the authors provide more detail about calculating the local potential and electric field as a function of cation concentration so we can better understand the slope in Fig 6e and 6f? Compare K^+ , Na^+ and Li^+ , we can see the slope of local potential for Li^+ is less positive than Na^+ and K^+ , and the slope of electric field for Li^+ is less negative than Na^+ and K^+ . Further, the absolute value of slope of local potential is "larger" than electric field for Li^+ , but the absolute value of slope of local potential is smaller than electric field for Na^+ and K^+ . This subtle difference can help explain why promotion effect surpasses suppression effect in Li^+ , but suppression effect outpaces promotion effect in Na^+ and K^+ (Fig 6g).

Minor comment

5. The authors labeled $K^+ > Na^+ > Li^+$ in Fig 2b. Yet their corresponding marks (effective cation size) on X-axis show that $K^+ < Na^+ < Li^+$. This is confusing. They may consider clarifying it in the figure caption of this panel.

Reviewer #2

(Remarks to the Author)

The submitted manuscript by Zhu et al. addresses a long-standing and important question in electrochemistry: the dual

promotional and inhibitory roles of alkali cations in alkaline hydrogen evolution reaction (HER). The authors propose a refined electric double layer (EDL) model that distinguishes between specifically adsorbed cations at the inner Helmholtz plane and solvated cations in the diffuse layer, and they argue that these two populations exert opposing influences on HER kinetics via modulation of the interfacial electric field. Overall, this is a well-written and conceptually innovative manuscript that advances our mechanistic understanding of cation effects in electrocatalysis. I recommend its publication after addressing the following minor points.

1. The proposed model considers the different specific adsorption capability of alkaline cations using standard adsorption potentials (-1.65 V for Li⁺, -1.27 V for Na⁺, and -1.19 V for K⁺). How are these values determined? I don't see any discussion or supporting reference.
2. The proposed model requires multiple parameters for calculation as listed in Table 1. How robust are the conclusions to variations in these inputs?

Reviewer #3

(Remarks to the Author)

This manuscript addresses an important and timely topic, namely the role of cations in electrocatalysis and their seemingly opposing effects on double-layer structure and catalytic activity. The modeling framework is thoughtfully constructed and incorporates several physically motivated aspects that are interesting. However, a number of key assumptions and parameter choices remain insufficiently justified, and several aspects of the model require further clarification and validation before the conclusions can be considered truly robust. Clarifying the physical basis of model parameters and providing at least a limited sensitivity analysis are essential for establishing the robustness of the conclusions. I therefore recommend major revisions before the manuscript can be considered for publication. My main concerns are outlined below.

1) Definition and physical meaning of effective ion sizes: The authors note that the literature reports a wide range of values for the "effective ion size" and subsequently adopt values of 8 Å, 7 Å, and 6 Å for Li⁺, Na⁺, and K⁺, respectively. Since these values directly determine the thickness of the Outer Helmholtz Layer in the model, their physical interpretation is crucial. At present, it is unclear how the effective ion size is defined. For example, does it correspond to the position of the first minimum in the cation–water radial distribution function (i.e., the boundary of the first hydration shell), the second minimum, or some other structural criterion? The manuscript would benefit from a clear and explicit definition of this quantity. Moreover, the cited values appear to be derived from bulk hydration properties. It is not obvious that these effective sizes, and in particular the trend Li⁺ > Na⁺ > K⁺ remain valid in the vicinity of a charged interface, where hydration structure and ion-water coordination can differ substantially from bulk behavior. The authors should discuss whether there is experimental or simulation-based evidence supporting the transferability of these effective ion sizes to interfacial environments.

2) Choice of the charge number parameter ξ : The manuscript employs a uniform value of $\xi = 0.85$ for all cations, yet the rationale for this choice is not explained. It is unclear why ξ is assumed to be ion-independent, given that different cations exhibit distinct hydration structures and polarizabilities. The authors are encouraged to clarify the physical meaning of ξ in their model and justify the choice of a universal value. In addition, a brief sensitivity analysis, either qualitative or quantitative, would be highly valuable. How sensitive are the predicted ion distributions, surface potentials, or double-layer characteristics to the chosen value of ξ . If ξ has only a minor influence on the key results, this should be explicitly demonstrated and discussed.

3) Lateral interaction coefficient γ : Similar concerns apply to the lateral interaction coefficient γ . The manuscript does not provide sufficient justification for the chosen value, nor for treating γ as ion-independent. One would intuitively expect lateral interactions to vary between different ionic species, especially if hydration effects or short-range correlations are implicitly included. The authors should clarify whether γ is intended to represent purely electrostatic interactions or whether it also accounts for hydration-mediated effects. A discussion of why γ is assumed to be the same for Li⁺, Na⁺, and K⁺ would significantly improve the interpretation of the model.

4) Treatment of interfacial water availability: The model incorporates solvent effects through finite ion size and electric-field-dependent water dissociation, which are meaningful and well-motivated refinements. However, it appears not to explicitly consider variations in the availability or density of water at the interface. Ion-specific adsorption and interfacial structuring can substantially alter the local water environment, which in turn may influence HER kinetics. The authors should comment on whether differences in interfacial water abundance—arising from the distinct adsorption characteristics of Li⁺, Na⁺, and K⁺—could also contribute to the experimentally observed trends in HER activity. If such effects are not included in the current framework, this limitation should be acknowledged, and the feasibility of incorporating interfacial water depletion or enrichment into the model should be discussed.

5) Broader parametric and sensitivity analysis: In the Results section, the inversion effect is primarily attributed to the chosen effective ion sizes, with the Frumkin effect following the trend K⁺ > Na⁺ > Li⁺, while the HER activity exhibits the opposite trend. At the same time, several other key parameters such as ξ , γ , and the interfacial dielectric constants ϵ_1 and ϵ_2 are treated as ion-independent. This raises the question of whether the reported inversion is uniquely determined by ion size, or whether similar behavior could emerge from alternative combinations of parameters. The authors should comment on whether different parameter sets might reproduce comparable trends.

A systematic parametric study varying the effective ion size, ξ , γ , and the interfacial dielectric constants ϵ_1 and ϵ_2 would substantially strengthen the manuscript. Even a limited sensitivity analysis would help disentangle the relative importance of each parameter, identify potential interdependencies, and demonstrate that the observed inversion is robust rather than an

artifact of a specific parameter choice. Such an analysis would greatly enhance the tractability, generality, and broader impact and value of the proposed model.

Reviewer #4

(Remarks to the Author)

Version 1:

Reviewer comments:

Reviewer #1

(Remarks to the Author)

The authors have made substantial revision to address my comments. Now I have no more questions.

Reviewer #2

(Remarks to the Author)

The authors have well addressed my previous concerns in the revised manuscript. Now I recommend its publication in Nature Communications.

Reviewer #3

(Remarks to the Author)

The authors have substantially revised this manuscript according to the extensive comments of all reviewers. In my view this work is now ready for publications.

Reviewer #4

(Remarks to the Author)

Reviewer 1:

This work took hydrogen evolution reaction (HER) in alkaline media as a representative model system and revisited the electric double layer (EDL) model and electron-transfer kinetics by considering impact of cation position on local electric field. They revealed that cations electrostatically adsorbed in the diffuse layer and those specifically adsorbed at the inner Helmholtz plane would have modulate the HER kinetics in opposite way: the former enhances HER while the latter suppresses it. The Janus-faced cation effects disclosed by their theoretical model successfully explained the observed manifold of cation effects in HER on Pt and Au. Overall, this is a timely and well-crafted paper that provides new physical insight into cations' influences in EDL formed on negatively charged catalyst surface. Specific comments to be addressed before further consideration are as follows.

Response:

We appreciate the reviewer's positive assessment and thoughtful suggestions, which have greatly benefited us in improving the manuscript.

Change:

In what follows, we provide a point-by-point response to the comments raised by the reviewer, and explain the changes we have made in the manuscript.

1. Previous studies (e.g., Phys. Chem. Chem. Phys., 2024, 26, 18233–18243; Nature Energy 5, 891–899 (2020)) have reported that under alkaline conditions there can be a relatively high OH/O coverage on metal surfaces. If OH/O adsorption is considered, qualitatively, how would it affect the conclusions of the present work?

Response:

We thank the reviewer for this insightful comment. Indeed, under alkaline conditions, co-adsorption of OH/O and cations occurs on certain metal surfaces, particularly on oxophilic metals, introducing additional complexity for the interpretation of cation effects.

In our study, we therefore focused on Au, for which OH/O adsorption is negligible in the potential region for HER. Modeling and parameterization are thus constructed specifically for Au. We did not employ the detailed model for Pt, where OH/O adsorption becomes relevant; instead, a qualitative discussion is provided in Figure 7 (Page 17).

Consequently, while OH/O adsorption affects HER kinetics on Pt, it does not alter the central conclusions of our work regarding the cation-induced electrostatic effects, which are quantitatively evaluated for Au.

Within our framework, adsorbed hydroxide species can have two opposing effects on HER: (i) enhancing the interfacial electric field due to their residual negative charge (J. Phys. Chem. C 2016, 120, 25, 13587–13595), thereby promoting HER; (ii) blocking the active sites at high coverage, which inhibits HER. The competition between these two effects can qualitatively explain the volcano-type relationship between HER activity and OH binding strength reported in the literature (Nature Energy 5, 891–899 (2020)). Although this prediction is qualitative, it illustrates that the model can be extended to anion effects and highlights the generality of the proposed mechanism, while not excluding other factors reported in prior studies.

Change:

Page 18: We rephrased the discussion on the role of adsorbed OH/O

“Although this study focuses on cation effects, the same logic should be applicable to anion effects as well. For instance, a volcano-type relationship between HER activity and hydroxide binding strength has been reported for modified Pt catalysts.³¹ Within our framework, adsorbed hydroxide species have two opposing effects: (i) enhancing the interfacial electric field due to their residual negative charge,^{38,54} thereby promoting HER, an effect opposite to that of specifically adsorbed cations; (ii) blocking active sites at high coverage, which inhibits HER. The competition between these two effects provides a qualitative explanation for the observed volcano-type trend.”

2. At the beginning of the manuscript, the authors state: “We focus on the Volmer step, which is generally regarded as the rate-determining step (RDS) of alkaline HER for the catalysts that bind hydrogen weakly, such as Hg, Ga and Au.” However, Pt is known to bind hydrogen relatively strongly (ACS Energy Lett. 2023, 8, 657–665). Why does the improved model proposed in this work also successfully fit and explain the experimental results for Pt, a metal with strong hydrogen adsorption?

Response:

As noted in our response to Comment 1, quantitative analyses conducted in this study have focused on Au, for which hydrogen binding is weak and the Volmer step is regarded as rate-determining. All model assumptions, parameterization, and evaluations are therefore conducted specifically for Au. We do not attempt to build a detailed kinetic model for Pt, for which hydrogen binds strongly and additional surface processes become relevant. However,

the water dissociation kinetics emphasized in this study is still rate-determining on Pt in alkaline solutions (Nature Energy 5, 891–899 (2020)).

The comparison to Pt shown in Figure 7 (Page 17) is intended to be illustrative rather than quantitative. In this figure, by adjusting only the adsorption strength of K^+ , which is stronger on Pt, the model reproduces several experimentally observed trends in cation concentration effects. This indicates that the mechanism proposed here is significant also on Pt.

However, HER on Pt involves additional complexities that are not captured by the present model, including strong hydrogen adsorption, and co-adsorption of hydroxide species. Therefore, while the model qualitatively captures certain trends, it does not fully reproduce the experimental data for Pt (J. Am. Chem. Soc. 2024, 146, 11, 7305–7312).

Accordingly, the agreement with selected experimental trends on Pt should be viewed as supportive evidence that electrostatic cation effects are also relevant on Pt. However, other mechanisms proposed in prior studies are not excluded.

Change:

Page 17: We clarified the limitations for Pt

“These varying trends reproduce the complex experimental behaviors in Figure 7a, where cation adsorption is stronger on Pt and at higher pH.^{22,76} We note, however, that HER on Pt involves additional complexities, including strong hydrogen adsorption and co-adsorption of hydroxide species.^{22,24,78} Thus, the qualitative comparison in Figure 7 indicates that the proposed mechanism remains relevant on Pt, but it does not exclude other effects reported in previous studies.¹⁵”

3. In the approximation treatment, two parameters— ψ_x (the local potential at the reaction plane, where a more positive ψ is expected to enhance HER in alkaline solutions) and the electric field—are considered separately. If the author can prove that the assumption holds in realistic conditions? If these two variables are associated, it would be necessary to discuss to what extent this approximation of treating them independently may affect the results.

Response:

ψ_x and $|\vec{E}_{el}|$ are not assumed to be independent variables in our work. They are obtained self-consistently from the solution of the EDL model (Eq. 2 and Eq. 3) and coupled via the free surface charge density, σ_{free} .

Specifically, for a given set of system parameters, σ_{free} is obtained by solving the EDL equations. Once σ_{free} is determined, the local potentials and electric field are calculated as:

$$\psi_{\text{OHP}} = \text{sign}(\sigma_{\text{free}}) \frac{2RT}{F} \text{arsinh} \left(\sqrt{\frac{1}{2\gamma} \left(\exp \left(\frac{\gamma}{2} \left(\frac{F\lambda_{\text{D}}\sigma_{\text{free}}}{RT\epsilon_{\text{S}}} \right)^2 \right) - 1 \right)} \right),$$

$$\psi_{\text{IHP}} = \psi_{\text{OHP}} + \frac{\sigma_{\text{free}}}{\epsilon_2} * \delta_2.$$

and

$$|\vec{E}_{\text{el}}| = |\sigma_{\text{free}}|/\epsilon_2.$$

Eq. 7 reflects the distinct roles of ψ_x and $|\vec{E}_{\text{el}}|$ in modulating the activation barrier. This formulation allows us to quantify the relative contributions of local potential and electric field to HER kinetics. Importantly, both effects evolve consistently with changes in electrolyte composition through the underlying EDL solution.

Change:

To facilitate transparency and reproducibility, all MATLAB codes used to solve the models and generate the figures are provided in the Supporting Information.

4. Can the authors provide more detail about calculating the local potential and electric field as a function of cation concentration so we can better understand the slope in Fig 6e and 6f? Compare K+, Na+ and Li+, we can see the slope of local potential for Li+ is less positive than Na+ and K+, and the slope of electric field for Li+ is less negative than Na+ and K+. Further, the absolute value of slope of local potential is "larger" than electric field for Li+, but the absolute value of slope of local potential is smaller than electric field for Na+ and K+. This subtle difference can help explain why promotion effect surpasses suppression effect in Li+, but suppression effect outpaces promotion effect in Na+ and K+ (Fig 6g).

Response:

To calculate the local potential and electric field as a function of cation concentration c_i , we first solve the cation adsorption coverage θ_i using Eq. 4, which depends on c_i . Using the resulting θ_i and other parameters listed in Table 1, the free surface charge density, σ_{free} , is obtained by solving the EDL model (Eq. 3). Once σ_{free} is obtained, the local potentials ψ_x and electric field strength $|\vec{E}_{\text{el}}|$ can be calculated, as described in the response to Comment 3.

This framework naturally explains the distinct slopes observed in Figure 6e and 6f. For Li+, the weaker adsorption and more gradual increase in θ_i with c_i lead to a smaller magnitude of

electric field variation. At the same time, the change in ψ_x remains comparatively significant. As a result, the promotion effect associated with ψ_x dominates for Li^+ . In contrast, for Na^+ and K^+ , stronger adsorption and a steeper increase in θ_i with c_i produce larger changes in $|\vec{E}_{el}|$. Consequently, the suppression effect associated with the electric field outweighs the promotion effect of ψ_x , as seen in Figure 6g.

For transparency and reproducibility, all MATLAB codes used to generate the main figures, including the local potential and electric field calculations, are provided in the Supporting Information.

Change:

Page 21: We added a new section “Numerical solution”

“Numerical solution

The models are solved using MATLAB, and all codes are provided in the SI. Using these codes, the main figures can be reproduced.”

5. The authors labeled $\text{K}^+ > \text{Na}^+ > \text{Li}^+$ in Fig 2b. Yet their corresponding marks (effective cation size) on X-axis show that $\text{K}^+ < \text{Na}^+ < \text{Li}^+$. This is confusing. They may consider clarifying it in the figure caption of this panel.

Response:

We thank the reviewer for pointing this out, and the figure caption has been revised to clarify the ordering of effective cation size.

Reviewer 2:

The submitted manuscript by Zhu et al. addresses a long-standing and important question in electrochemistry: the dual promotional and inhibitory roles of alkali cations in alkaline hydrogen evolution reaction (HER). The authors propose a refined electric double layer (EDL) model that distinguishes between specifically adsorbed cations at the inner Helmholtz plane and solvated cations in the diffuse layer, and they argue that these two populations exert opposing influences on HER kinetics via modulation of the interfacial electric field. Overall, this is a well-written and conceptually innovative manuscript that advances our mechanistic understanding of cation effects in electrocatalysis. I recommend its publication after addressing the following minor points.

Response:

We sincerely thank the reviewer for the positive evaluation and thoughtful comments. We are encouraged by the recognition of the conceptual novelty of our work.

Change:

In what follows, we provide a point-by-point response to the comments raised by the reviewer, and explain the changes we have made in the manuscript.

1. The proposed model considers the different specific adsorption capability of alkaline cations using standard adsorption potentials (-1.65 V for Li⁺, -1.27 V for Na⁺, and -1.19 V for K⁺). How are these values determined? I don't see any discussion or supporting reference.

Response:

We thank the reviewer for pointing this out. In our model, the equilibrium adsorption potential is an effective parameter that captures relative cation adsorption strength, which follows the order Li⁺ < Na⁺ < K⁺ (Nat. Catal. 5, 624–632, 2022; Can. J. Chem. 59, 1944–1953, 1981). Accordingly, we adopt the criterion $E_{\text{Li}^+}^0 < E_{\text{Na}^+}^0 < E_{\text{K}^+}^0$. The numerical values $E_{\text{Li}^+}^0 = -1.65$ V, $E_{\text{Na}^+}^0 = -1.27$ V, and $E_{\text{K}^+}^0 = -1.19$ V are obtained by fitting to the experimental polarization curves in Figure 5b. The validity of these values is supported by the results in Figure 6, where the same set of parameters reproduces also the experimental trends for cation concentration effects.

We note that E_i^0 is a key parameter that could depend on catalyst material and solution pH. To assess its influence, we performed a systematic analysis shown in Figure 7. The results indicate that several observed cation trends can be reproduced with different values of E_i^0 .

This supports the key mechanism of this work, which emphasizes the relative adsorption strength of cations.

Change:

Page 21:

“The cation adsorption ability follows the trend $K^+ > Na^+ > Li^+$,^{52,53} implying $E_{Li^+}^0 < E_{Na^+}^0 < E_{K^+}^0$. We obtained $E_{Li^+}^0 = -1.65$ V, $E_{Na^+}^0 = -1.27$ V, and $E_{K^+}^0 = -1.19$ V versus SHE by fitting to experimental polarization curves. The validity of these values is supported by the results in Figure 6, where the same set of parameters reproduces also the experimental trends for cation concentration effects.”

2. The proposed model requires multiple parameters for calculation as listed in Table 1. How robust are the conclusions to variations in these inputs?

Response:

We thank the reviewer for raising this important question. Although the model contains several parameters, the inversion of cation trends in our model is governed primarily by two well-established cation properties: effective cation size ($Li^+ > Na^+ > K^+$) and adsorption strength ($Li^+ < Na^+ < K^+$), represented in the model by the trends $d_{Li^+} > d_{Na^+} > d_{K^+}$ and $E_{Li^+}^0 < E_{Na^+}^0 < E_{K^+}^0$. These two parameters control the Frumkin effect and the electric-field effect, whose competition constitutes the central physical picture proposed in this work.

All remaining parameters were treated as ion-independent. This modeling strategy minimizes the number of adjustable parameters, isolates the consequences of the two physically motivated cation trends, and thereby enhances the robustness and interpretability of the model.

To further assess the robustness of conclusions, we performed additional sensitivity analyses in which the parameters that could plausibly be cation-dependent, including ξ_i , γ_i , and the interfacial dielectric constants ϵ_1 and ϵ_2 , were independently varied over physically reasonable ranges. As shown and discussed in Supplementary Note 1, these variations modify the electric double layer characteristics and the HER current density, but do not reproduce the inversions of cation trends without accounting for the systematic trends in effective cation size and adsorption strength.

We therefore conclude that the reported inversion is a robust and generic consequence of the competition between the Frumkin effect and the electric-field effect, both of which are fundamentally governed by effective cation size and adsorption strength. This point has been

clarified in the revised manuscript, and a new subsection entitled “Parameter sensitivity analysis” has been added to the “Methods” section, with the corresponding results presented in the Supplementary Note 1.

Change:

Page 21:

We added a new subsection “Parameter sensitivity analysis” to the “Methods” section to systematically examine the robustness of the model with respect to variations in key input parameters.

Supporting Information:

We added a new section “Supplementary Note 1: Parameter sensitivity analysis”.

Reviewer 3:

This manuscript addresses an important and timely topic, namely the role of cations in electrocatalysis and their seemingly opposing effects on double-layer structure and catalytic activity. The modeling framework is thoughtfully constructed and incorporates several physically motivated aspects that are interesting. However, a number of key assumptions and parameter choices remain insufficiently justified, and several aspects of the model require further clarification and validation before the conclusions can be considered truly robust. Clarifying the physical basis of model parameters and providing at least a limited sensitivity analysis are essential for establishing the robustness of the conclusions. I therefore recommend major revisions before the manuscript can be considered for publication. My main concerns are outlined below.

Response:

We are grateful to the reviewer for a careful reading of the manuscript, and many comments that lead to substantial improvement of the work and the manuscript.

Change:

In what follows, we provide a point-by-point response to the comments raised by the reviewer, and explain the changes we have made in the manuscript.

1. Definition and physical meaning of effective ion sizes: The authors note that the literature reports a wide range of values for the “effective ion size” and subsequently adopt values of 8 Å, 7 Å, and 6 Å for Li⁺, Na⁺, and K⁺, respectively. Since these values directly determine the thickness of the Outer Helmholtz Layer in the model, their physical interpretation is crucial. At present, it is unclear how the effective ion size is defined. For example, does it correspond to the position of the first minimum in the cation–water radial distribution function (i.e., the boundary of the first hydration shell), the second minimum, or some other structural criterion? The manuscript would benefit from a clear and explicit definition of this quantity. Moreover, the cited values appear to be derived from bulk hydration properties. It is not obvious that these effective sizes, and in particular the trend Li⁺ > Na⁺ > K⁺ remain valid in the vicinity of a charged interface, where hydration structure and ion-water coordination can differ substantially from bulk behavior. The authors should discuss whether there is experimental or simulation-based evidence supporting the transferability of these effective ion sizes to interfacial environments.

Response:

We agree with the reviewer that a clear definition of this parameter is essential. In the Bikerman model employed in our work, the effective ion size does not represent a uniquely defined microscopic ionic radius (e.g., the radial distribution function minimum), but rather the lattice cell size that determines the excluded-volume entropy and the maximal ion concentration. Accordingly, this phenomenological parameter should be interpreted as an exclusion length relevant for steric saturation.

Within this approach, ions with a larger effective size exhibit a reduced tendency to accumulate near the surface, leading to a lower electric double layer (EDL) capacitance (J. Phys. Chem. B 2007, 111, 20, 5545–5557). Smaller bare cations tend to be more strongly solvated and therefore possess larger effective sizes (J. Phys. Chem. 1959, 63, 9, 1381–1387), inducing the trend $d_{\text{Li}^+} > d_{\text{Na}^+} > d_{\text{K}^+}$ adopted here. This ordering is consistent with experimental observations showing larger EDL capacitance in K^+ -containing electrolytes compared to Na^+ - and Li^+ -containing ones (J. Phys. Chem. Lett. 2018, 9, 8, 1927–1930; Energy Environ. Sci., 2019, 12, 3001-3014).

Importantly, the transferability of this trend to charged interfaces is supported by first-principles calculations indicating that cation concentration at the outer Helmholtz plane follows $\text{K}^+ > \text{Na}^+ > \text{Li}^+$ (Nat Catal 4, 654–662, 2021; J. Am. Chem. Soc. 2017, 139, 32, 11277–11287), as well as by semiclassical modeling of the capacitance on Hg (JACS Au 2025, 5, 7, 3453–3467). Combined spectroscopic and simulation studies reveal stronger interfacial electric fields in K^+ electrolytes than in Li^+ electrolytes (ChemRxiv 2024, Doi: 10.26434/chemrxiv-2024-hrnb5; Energy Environ. Sci., 2019, 12, 3001-3014), which is consistent with enhanced K^+ accumulation relative to Li^+ near negatively charged surfaces.

The thickness of the outer Helmholtz layer (OHL) influences the Helmholtz capacitance, $C_{\text{H}} = \frac{1}{\frac{\delta_1 + \delta_2}{\epsilon_1} + \frac{\delta_2}{\epsilon_2}}$, with δ_2 being the OHL thickness. Using effective sizes 8 Å, 7 Å, and 6 Å for Li^+ , Na^+ , and K^+ , yields the trend $C_{\text{H}}(\text{K}^+) > C_{\text{H}}(\text{Na}^+) > C_{\text{H}}(\text{Li}^+)$, while the relative differences remain small ($\text{Li}^+/\text{Na}^+ = 96\%$ and $\text{Na}^+/\text{K}^+ = 95\%$), of the same order as the capacitance measurements by Grahame (J. Electrochem. Soc. 98, 343, 1951). In addition, the distances between the cations and the electrode surface ($\delta_1 + \delta_2$) lie between 5 Å and 6 Å, consistent with molecular dynamics simulations (Chem. Phys. Lett. 2022, 795, 139518; J. Am. Chem. Soc. 2023, 145, 3, 1897–1905).

Taken together, these results support the use of a larger effective size for more strongly hydrated cations, i.e., $d_{\text{Li}^+} > d_{\text{Na}^+} > d_{\text{K}^+}$. This trend has also been assigned as a main mechanism for understanding cation effects in a recent review (Nat Catal 8, 986–999, 2025). While the absolute values are phenomenological and could depend on interfacial conditions, the qualitative trends in interfacial structure and HER activity reported in Figure 2 are robust with respect to variations in the effective ion size.

Change:

We provided a clarification for the definition and trends of effective cation size.

Page 7:

“where d_i is the effective ion size, defined as a phenomenological lattice cell size that determines the excluded-volume entropy and the maximum local ion concentration.

Cations are distinguished by their effective sizes. Although reported values vary considerably across different sources, the relative trend is generally consistent, following the order $d_{\text{Li}^+} > d_{\text{Na}^+} > d_{\text{K}^+}$, due to the stronger solvation of Li^+ compared to Na^+ and K^+ .^{9,13,40–43} Importantly, this trend is preserved at charged interfaces, as supported by first-principles calculations,^{10,44} semiclassical modeling,⁴⁵ and spectroscopic studies.^{9,46}”

Page 20:

“The resulting distances between the cations and the electrode surface, *i.e.*, $\delta_1 + \delta_2$, lie between 5 Å and 6 Å, consistent with molecular dynamic simulations.^{85,86}”

2. Choice of the charge number parameter ξ : The manuscript employs a uniform value of $\xi = 0.85$ for all cations, yet the rationale for this choice is not explained. It is unclear why ξ is assumed to be ion-independent, given that different cations exhibit distinct hydration structures and polarizabilities. The authors are encouraged to clarify the physical meaning of ξ in their model and justify the choice of a universal value. In addition, a brief sensitivity analysis, either qualitative or quantitative, would be highly valuable. How sensitive are the predicted ion distributions, surface potentials, or double-layer characteristics to the chosen value of ξ . If ξ has only a minor influence on the key results, this should be explicitly demonstrated and discussed.

Response:

In the modified electric double layer (EDL) model accounting for partially charged adsorbates (Eq. 3), ξ_i represents the charge number per adsorbed cation. It reflects partial electronic charge transfer between the adsorbed cation and the electrode, as well as electrostatic screening at the interface (JACS Au 2023, 3, 2, 550–564). In principle, this parameter can be estimated from first-principles calculations, for example via Bader charge analysis.

Recent AIMD studies combined with Bader charge analysis indicate that ξ_i is only weakly dependent on cation identity for alkali metal cations, with reported values typically lying in the range 0.8–0.9 (Nat Catal 4, 654–662, 2021; Nat Catal 5, 624–632, 2022; J. Am. Chem. Soc. 2023, 145, 3, 1897–1905). Based on these results, we adopted $\xi_i = 0.85$ for all cations. This choice is consistent with AIMD simulations, and reduces the number of adjustable parameters in the

model, thereby avoiding overfitting and allowing the analysis to focus on the key cation-specific factors of interest.

We assessed the sensitivity of model results to ξ_i by performing a sensitivity analysis, in which ξ_i was varied over a reasonable range. The effects on cation adsorption coverage, EDL characteristics (ψ_x and $|\vec{E}_{el}|$), and the HER current density are shown in Figure S5.

Change:

Page 10:

“Here, ξ_i is the charge number per adsorbed cation and can be estimated from , e.g., Bader charge analysis.”

Page 20:

“Recent AIMD studies combined with Bader charge analysis indicate that ξ_i is only weakly dependent on cation identity, with typical values obtained in the range 0.8-0.9 for alkali metal cations.^{44,52,86} We thus adopt $\xi_i = 0.85$ for all cations. The sensitivity of model results to ξ_i has been assessed by performing a sensitivity analysis as reported in the SI and illustrated in Figure S5.”

Supporting Information (Figure S5):

Figure S5: Influence of ξ_i on (a) Cation coverage, (b) ψ_x (assumed $\psi_x = \psi_{\text{OHP}}$), (c) $|\vec{E}_{\text{el}}|$, and (d) HER current density. Conditions: $c_b = 0.1$ M and pH = 13.

“ ξ_i influences the cation adsorption isotherm through Eq. 4. A smaller ξ_i corresponds to a larger degree of partial charge transfer, leading to a steeper increase in cation adsorption with increasingly negative potential, as shown in Figure S5a. In addition, ξ_i affects the EDL characteristics through the surface charge contribution $\sigma_{\text{ad}} = e_0 \rho \xi_i \theta_i$ in Eq. 3, which modifies both the local potential and the electric field strength, as illustrates in Figure S5b and S5c. While variations in ξ_i rescale the magnitude of the HER current density, they do not reproduce the experimentally observed inversion of cation trends, Figure S5d.”

3. Lateral interaction coefficient γ : Similar concerns apply to the lateral interaction coefficient γ . The manuscript does not provide sufficient justification for the chosen value, nor for treating γ as ion-independent. One would intuitively expect lateral interactions to vary between different ionic species, especially if hydration effects or short-range correlations are implicitly included. The authors should clarify whether γ is intended to represent purely electrostatic interactions or whether it also accounts for hydration-mediated effects. A discussion of why γ is assumed to be the same for Li^+ , Na^+ , and K^+ would significantly improve the interpretation of the model.

Response:

We describe the cation adsorption using a Frumkin isotherm, which includes a simplified, mean-field treatment of lateral interactions. Within this framework, γ_i is a phenomenological parameter that accounts for the effective repulsion between adsorbed cations, arising from a combination of electrostatic interactions and short-range correlations mediated by water and the catalyst surface.

In this study, we adopt a cation-independent γ_i for three main reasons. Firstly, for dehydrated cations that adsorb on the catalyst surface with a low coverage ($\theta_i < 0.18$ in our model, as shown in Figure S2), we expect that the dominant repulsion originates from electrostatic interactions, which should depend only weakly on cation identity, whereas the hydration-mediated effects and short-range correlations become more relevant at high coverage. Secondly, introducing cation-specific γ_i values would add additional uncertain parameters to the model, thereby reducing its robustness. Thirdly, cation specificity in our framework is captured through two physically justified parameters with established trends, i.e., the effective cation size and the equilibrium adsorption potential, while all other parameters are kept identical across cations. This strategy maintains a minimal and physically interpretable parameter set and avoids overfitting.

We acknowledge that a fully self-consistent, cation-specific treatment would be valuable, but this requires detailed microscopic inputs from experiments and first-principles calculations, which lie beyond the scope of this work. As noted on page 10:

“We note that this is a simplified consideration, and a more comprehensive description would require accounting for the complex interactions between cations, water molecules, and the catalyst surface, and would relax the mean-field description.”

To assess the role of γ_i , we performed a sensitivity analysis in Figure S6. While a variation in γ_i modifies the cation adsorption, EDL characteristics, and HER kinetics, it does not reproduce the experimentally observed inversion of cation trends without accounting for the systematic trends in effective cation size and adsorption strength.

Change:

Supporting Information (Figure S6):

Figure S6: Influence of γ_i on (a) Cation coverage, (b) ψ_x (assumed $\psi_x = \psi_{\text{OHP}}$), (c) $|\vec{E}_{\text{el}}|$, and (d) HER current density. Conditions: $c_b = 0.1$ M and $\text{pH} = 13$.

“ γ_i represents the lateral repulsion between adsorbed cations in the Frumkin isotherm in Eq. 4. A larger γ_i increases the effective repulsive interaction, thereby inhibiting cation adsorption,

as shown in Figure S6a. As a consequence, ψ_x is shifted to more negative values, while \vec{E}_{el} is enhanced with increasing γ_i , as shown in Figure S6b and S6c. The competition between these two effects leads to an overall increase in the HER current density with increasing γ_i , Figure S6d. Importantly, varying γ_i alone does not reproduce the experimentally observed inversion of cation trends.”

4. Treatment of interfacial water availability: The model incorporates solvent effects through finite ion size and electric-field-dependent water dissociation, which are meaningful and well-motivated refinements. However, it appears not to explicitly consider variations in the availability or density of water at the interface. Ion-specific adsorption and interfacial structuring can substantially alter the local water environment, which in turn may influence HER kinetics. The authors should comment on whether differences in interfacial water abundance—arising from the distinct adsorption characteristics of Li^+ , Na^+ , and K^+ —could also contribute to the experimentally observed trends in HER activity. If such effects are not included in the current framework, this limitation should be acknowledged, and the feasibility of incorporating interfacial water depletion or enrichment into the model should be discussed.

Response:

We thank the reviewer for this insightful comment. In the previous framework, the availability of interfacial water was treated as constant. To relax this simplification, we consider a scenario in which the interfacial water density is proportional to the unoccupied surface fraction, $(1 - \theta_i)$, where θ_i denotes the adsorption coverage of Li^+ , Na^+ , or K^+ . This treatment provides an estimate of interfacial water depletion arising from ion-specific adsorption and naturally incorporates dependencies on both potential and cation identity. Notably, a reduction in first-layer water availability is observed only at very negative potentials ($E_M - E_{pzc} < -1.89$ V) in AIMD simulations (Nat. Mater. 18, 697–701, 2019), which is consistent with the potential range where significant cation adsorption occurs in our model ($E_M < -0.6$ V vs RHE in Figure S2).

Accordingly, Eq. 8 for the current density becomes

$$j = -2\kappa_{el}v_n \exp\left(-\frac{\Delta G_a}{k_B T}\right) e_0 \rho (1 - \theta_i).$$

Using this expression, we have recalculated the HER current densities and Tafel slopes presented in Figures 5-7 and S3. The results indicate that while the absolute magnitude of the HER current is slightly adjusted, the qualitative trends remain unchanged. In particular, the relative ordering of HER activity among Li^+ , Na^+ , and K^+ , as well as the competition between

the Frumkin effect and the electric-field effect, are preserved. For the reviewers' convenience, we show the key comparisons here.

These results further support the central conclusion of this work, namely that cation positioning and its influence on local electrostatics constitute the dominant factors governing the observed cation effects.

Change:

Page 12: Eq. 8 is updated to $j = -2\kappa_{el}v_n \exp\left(-\frac{\Delta G_a}{k_B T}\right) e_0 \rho(1 - \theta_i)$.

Figures 5-7 and S3 have been recalculated and updated using this revised expression.

5. Broader parametric and sensitivity analysis: In the Results section, the inversion effect is primarily attributed to the chosen effective ion sizes, with the Frumkin effect following the trend $K^+ > Na^+ > Li^+$, while the HER activity exhibits the opposite trend. At the same time, several other key parameters such as ξ , γ , and the interfacial dielectric constants ϵ_1 and ϵ_2 are treated as ion-independent. This raises the question of whether the reported inversion is uniquely determined

by ion size, or whether similar behavior could emerge from alternative combinations of parameters. The authors should comment on whether different parameter sets might reproduce comparable trends. A systematic parametric study varying the effective ion size, ξ , γ , and the interfacial dielectric constants ϵ_1 and ϵ_2 would substantially strengthen the manuscript. Even a limited sensitivity analysis would help disentangle the relative importance of each parameter, identify potential interdependencies, and demonstrate that the observed inversion is robust rather than an artifact of a specific parameter choice. Such an analysis would greatly enhance the tractability, generality, and broader impact and value of the proposed model.

Response:

We thank the reviewer for this comprehensive and insightful comment regarding parameter sensitivity. From a purely mathematical perspective, alternative parameter sets may exist that reproduce similar qualitative trends. However, from a physical standpoint, the inversion of cation trends in our model is governed primarily by two well-established cation properties: the effective cation size ($\text{Li}^+ > \text{Na}^+ > \text{K}^+$) and the adsorption strength ($\text{Li}^+ < \text{Na}^+ < \text{K}^+$). The interplay of these properties constitutes the central physical picture proposed in our work.

Other parameters, including ξ_i , γ_i , and the interfacial dielectric constants ϵ_1 and ϵ_2 , were treated as ion-independent. This choice is made deliberately to isolate the consequences of the two physically justified cation trends and to avoid introducing additional, poorly constrained degrees of freedom that could obscure the underlying mechanism.

To address the reviewer's concern more directly, we have performed additional sensitivity analyses, in which ξ_i , γ_i , ϵ_1 and ϵ_2 are independently varied over physically reasonable ranges. These results, now included and discussed in the Supporting Information, demonstrate that the experimentally observed inversion of cation trends cannot be reproduced by varying these parameters alone, i.e., without accounting for the systematic trends in effective cation size and adsorption strength.

We therefore conclude that the reported inversion is not an artifact of particular parameter choices, but rather a robust and generic consequence of the competition between the Frumkin effect and the electric-field effect, both of which are ultimately governed by effective cation size and adsorption strength. This point has been clarified in the revised manuscript, and a new subsection entitled "Parameter sensitivity analysis" has been added to the "Methods" section, with the corresponding results presented in the Supporting Information.

Change:

We added a new section "Parameter sensitivity analysis" on Page 21:

“Cation specificity in our framework is captured through two physically motivated parameters with established trends, *i.e.*, the effective size, d_i , and the equilibrium cation adsorption potential, E_i^0 , while all other parameters are kept identical for different cations. This strategy maintains a minimal and physically interpretable parameter set and avoids overfitting.

To disentangle the influence of other parameters, including ξ_i , γ_i , ϵ_1 and ϵ_2 , we performed sensitivity analyses, in which these parameters were independently varied over reasonable ranges. The impact of these variations on key observables, including cation adsorption coverage, EDL characteristics (ψ_x and $|\vec{E}_{el}|$), and HER current density, has been systematically analyzed.

The results, summarized and discussed in the SI, demonstrate that variations in these parameters affect the magnitude of interfacial potential, electric field strength, and HER current density, but do not reproduce the experimentally observed inversion of cation trends without accounting for the systematic trends in effective cation size and adsorption strength. This supports the proposed mechanism that attributes the observed inversions of cation effects to the competition between Frumkin effect and electric field effect, both of which are ultimately governed by cation size and adsorption strength.

While we believe that the conclusions are well justified and robust, we expect our work to stimulate more detailed simulations at the level of DFT and AIMD to systematically scrutinize the impact of different cation states on the HER kinetics.”

We added “Supplementary Note 1: Parameter sensitivity analysis” in the Supporting Information, in which the sensitivity of the model to the parameters ξ_i , γ_i , ϵ_1 and ϵ_2 is examined in detail.

We provided all MATLAB codes used to generate the main figures in the Supporting Information for transparency and reproducibility.

Reviewer 4:

Response:

We sincerely thank the reviewer for co-reviewing our manuscript and for the thoughtful feedback, which has helped us improve the clarity and quality of our work.